# Assessing insecticide susceptibility, diagnostic dose and time for the sand fly *Phlebotomus argentipes*, the vector of visceral leishmaniasis in India, using the CDC bottle bioassay

Rahul Chaubey[1,2], Ashish Shukla[2], Anurag Kumar Kushwaha[2], Puja Tiwary[2], Shakti Kumar Singh[1], Shawna Hennings[3], Om Praksh Singh[4], Phillip Lawyer[5], Edgar Rowton[6], Christine A. Petersen[7,8], Scott A. Bernhardt[3]*, Shyam Sundar[2]

**1** Kala-Azar Medical Research Center, Muzaffarpur, Bihar, India, **2** Department of Medicine, Institute of Medical Sciences, Banaras Hindu University, Varanasi, India, **3** Department of Biology, Utah State University, Logan, Utah, United States of America, **4** Department of Biochemistry, Institute of Science, Banaras Hindu University, Varanasi, India, **5** Arthropod Collections, Monte L. Bean Life Science Museum, Brigham Young University, Provo, Utah, United States of America, **6** Division of Entomology, Walter Reed Army Institute of Research, Silver Spring, Maryland, United States of America, **7** Department of Epidemiology, College of Public Health, University of Iowa, Iowa City, Iowa, United States of America, **8** Center for Emerging Infectious Diseases, University of Iowa, Coralville, Iowa, United States of America

* scott.bernhardt@usu.edu

**Data Availability Statement:** The authors confirm that all data underlying the findings are fully

## Abstract

Visceral leishmaniasis (VL) is a vector-borne protozoan disease, which can be fatal if left untreated. Synthetic chemical insecticides are very effective tools for controlling of insect vectors, including the sand fly *Phlebotomus argentipes*, the vector of VL in the Indian subcontinent. However, repeated use of the same insecticide with increasing doses potentially can create high selection pressure and lead to tolerance and resistance development. The objective of this study was to determine the lethal concentrations and assess levels of susceptibility, diagnostic doses and times to death of laboratory-reared *P. argentipes* to five insecticides that are used worldwide to control vectors. Using the Center for Disease Control and Prevention (CDC) bottle bioassay, 20–30 sand flies were exposed in insecticide- coated 500-ml glass bottles. Flies were then observed for 24 hours and mortality was recorded. Dose-response survival curves were generated for each insecticide using QCal software and lethal concentrations causing 50%, 90% and 95% mortality were determined. A bioassay was also conducted to determine diagnostic doses and diagnostic times by exposing 20–30 flies in each bottle containing set concentrations of insecticide. Mortality was recorded at 10-minute intervals for 120 minutes to generate the survival curve. *Phlebotomus argentipes* are highly susceptible to alpha-cypermethrin, followed by deltamethrin, malathion, chlorpyrifos, and least susceptible to DDT. Also, the lowest diagnostic doses and diagnostic times were established for alpha-cypermethrin (3μg/ml for 40 minutes) to kill 100% of the flies. The susceptibility data, diagnostic doses and diagnostic times presented here will be useful as baseline reference points for future studies to assess insecticide susceptibility and resistance monitoring of field caught sand flies and to assist in surveillance as VL elimination is achieved in the region.

available without restriction. All relevant data are within the manuscript.

**Funding:** This work was supported by the Extramural Program of the National Institute of Allergy and Infectious Diseases, National Institutes of Health (Tropical Medicine Research Center grant U19AI074321 to SS). The funders had no role in study design, data collection and analysis, decision to publish or preparation of the manuscript.

**Competing interests:** The authors have declared that no competing interests exist.

## Author summary

Synthetic chemical insecticides are mainly used for controlling sand flies via indoor residual spraying (IRS) and insecticide treated nets. Sand flies had developed resistance against DDT in endemic areas of Bihar, India, and was replaced by synthetic pyrethroids (alpha-cypermethrin) in the second phase of IRS in 2015. The substantial increase in the use of synthetic pyrethroids will potentially lead to increased resistance because DDT and pyrethroids both have the same target site. To monitor insecticide resistance in field caught sand flies, susceptibility status and diagnostic doses and times are warranted, as no diagnostic dose for resistance detection has been established for Indian sand flies. A laboratory reared *P. argentipes* was used to quantify insecticide susceptibility and determination of diagnostic doses and times. The CDC bottle bioassay was used with five insecticides viz., alpha-cypermethrin, deltamethrin, malathion, chlorpyrifos and DDT. Results indicate that the *P. argentipes* were highly susceptible to alpha-cypermethrin followed by deltamethrin, malathion, chlorpyrifos and least susceptible to DDT. This study provides critical base line reference data for further studies to assess insecticide susceptibility and resistance monitoring in field populations, as well as determining diagnostic doses and times for other insecticide susceptible sand fly populations undergoing surveillance to sustain VL elimination.

## Introduction

Visceral Leishmaniasis (VL), known as kala azar in the Indian subcontinent, is a parasitic disease caused by *Leishmania donovani*, which is transmitted by the bite of an infected female sand fly *Phlebotomus argentipes* Annandale & Brunneti (Diptera: Psychodidae). *Phlebotomus argentipes* is the only proven vector in the Southeast Asia Region. Control of VL throughout the Indian subcontinent has relied on the use of synthetic insecticides through indoor residual spraying (IRS). The vector, *P. argentipes*, is tiny and fragile, a weak flyer, prefers to hop short distances, and rest in dark corners of houses, cattle sheds and other dwellings [1]. This resting behavior makes them a suitable target for control by IRS with insecticides. As a result, during the time period of 1953–1962, IRS performed by the Indian national malaria program, using DDT for malaria control, had an immense effect in decreasing sand fly populations and significantly reduced VL cases in the Indian subcontinent [2–5]. This led to the adoption of IRS by the Indian VL elimination program as the main focus for *P. argentipes* control. Visceral leishmaniasis resurged in the 1990s, and in 2003, India launched a kala-azar elimination program using DDT aimed at eliminating VL from the country by 2015 [6]. Given the paramount importance of IRS to the VL control effort, insecticide resistance poses a very real threat to achieving and sustaining the elimination goals. Due to the declining effectiveness of DDT in sand fly control, the synthetic pyrethroid alpha-cypermethrin (5% WP) was introduced as an alternative in the second phase of IRS [7].

Since their introduction, control of arthropod vectors via chemical insecticides are a key focus of control programs to mitigate transmission of vector borne infections [8]. Insecticide resistance is a pre-adaptive phenomenon and is generally considered one of the most serious obstacles to effective vector control. Unfortunately, indiscriminate use of insecticides exerts tremendous selective pressure for the development of insecticide resistance [9,10]. Increasing insecticide dosage only intensifies the problem of resistance by increasing the frequency of genetic traits in a vector population [9]. Metabolic-detoxification and target-site insensitivity

are the two mechanisms of resistance observed in all classes of insecticides in all the major vector species [11,12]. Acquiring data on the susceptibility to insecticides and their diagnostic doses and diagnostic times will support and direct the strategy of effective vector management programs.

The WHO exposure-kit bioassay and the Centers for Disease Control and Prevention (CDC) bottle bioassay are the two techniques most commonly used to measure a vector species susceptibility to insecticides [13,14]. The WHO exposure-kit bioassay is widely accepted because it can measure insecticide susceptibility in many species of insect vectors worldwide [15–19]. The assays can be run with live insects collected in the field or with their progeny reared in the laboratory. The WHO bioassay is a standardized protocol that consists of an exposure kit containing tubes lined with filter papers impregnated with a specific concentration of an insecticide [14,20]. Despite its accepted use, the WHO bioassay is expensive, filter papers are not available for some insecticides, and there is a limited range of concentrations that can be purchased for some insecticides [18,21]. The CDC bottle bioassay is an economical and portable alternative to the WHO exposure-kit bioassay [14,18,21] with materials that can be acquired locally and prepared in field locations [22].

Sand flies are among the insect vectors that require resistance monitoring because they have been continuously targeted with different classes of insecticides such as organochlorines, organophosphates, carbamates, and pyrethroids via residual spraying, ultra-low volume spraying, insecticide-treated clothing, and insecticide-treated nets [23–27]. These exposures are either in focused vector control efforts or unintentional as part of vector control efforts targeting other vector species. In the late 1970s, *P. argentipes* was understood to be susceptible to DDT, but since then there is much evidence of its resistance to DDT throughout the endemic districts of Bihar, India [4,27–30]. Various studies on Indian sand fly populations, as well as global populations of sand flies, demonstrate that *P. argentipes* is either susceptible, tolerant or resistant to different classes of insecticides using the WHO exposure kit bioassay [1,24,25,27,31]. However, studies from other parts of the globe using the CDC bottle bioassay to assess the susceptibility status [25,26,32,33] and effective diagnostic doses and times for sand fly populations to insecticides are very limited [32,34,35]. Insecticide susceptibility data including diagnostic doses and diagnostic times are limited for Indians and fly populations using the CDC bottle bioassay.

The aim of this study was to quantify, using the CDC bottle bioassay, the susceptibility of laboratory reared *P. argentipes* and determine the diagnostic doses and times to five insecticides viz. alpha-cypermethrin, deltamethrin, chlorpyrifos, malathion and DDT. A dose-response survival curve was produced for each insecticide and from each curve, $LC_{50}$, $LC_{90}$, and $LC_{95}$ values were determined, as well as the diagnostic doses and diagnostic times for the same. These doses can now be used for comparison in future studies to assess *P. argentipes* susceptibility and monitor insecticide resistance in field caught sand flies to assist in surveillance. As VL elimination is achieved in the region and continued efforts and vigilance required to sustain the gain and maintain the validation of elimination once achieved.

## Material and methods

### Ethics statement

This work was conducted with ethical approval (Letter No.-CAEC/DEAN/2014/CAEC/615) obtained from Institutional Review Committees of Banaras Hindu University, Varanasi, India and Kala-azar Medical Research Centre (KAMRC), Muzaffarpur, India and University of Iowa, Institutional Animal Care and Use Committee (IACUC) protocol number 9041721.

## Sand flies

Laboratory-reared populations of *P. argentipes* were obtained from an established closed colony at the Kala-Azar Medical Research Centre (KAMRC), Muzaffarpur, Bihar [36]. Wild sand flies were collected from the selected village (after coordinating with ministry of Health IRS teams) in which residual spraying had not been done recently (within 1 or 2 years). From March through mid-December 2015, over 68,000 sand flies were collected from human dwellings and cattle sheds using CDC-type light traps over 254 nights. Blood-fed and gravid *P. argentipes* females were aspirated from collection bags and placed individually in isoline-rearing vials for oviposition. More than 2,500 egg clutches were harvested and reared according to standard methods, providing a continuous critical mass of F1 males and females to stimulate social feeding behavior. Once the colony became self-sustaining, it was closed to infusion with wild-caught material and certified free of specific human pathogens. The closed colony has never been exposed to any insecticides, even after the 20[th] generation with the start of the susceptibility study.

## Insecticides

Five technical-grade insecticides were used in this study: two pyrethroids [alpha-cypermethrin (Sigma- Aldrich) and deltamethrin (Sigma- Aldrich)]; two organophosphates [chlorpyrifos (Chem Service) and malathion (Chem Service)], and the organochlorine [dichlorodiphenyltrichloroethane (DDT) (Agilent Technologies)]. All insecticide dilutions were prepared in acetone, stored in glass bottles wrapped in aluminum foil and kept at 4°C when not in use [35]. The concentrations of each insecticide to which *P. argentipes* was exposed are listed in Table 1. The values of lethal concentrations causing 50% and 90% mortality for each insecticide were used for determining the diagnostic doses and times.

## Preparation of exposure bottle

The day prior to exposing the sand flies, 500-ml glass bottles were prepared by coating them on the inside with the designated insecticides as described by Denlinger et al [33]. Insecticide concentrations in different sized bottles were calculated according to the CDC method developed by Brogdon and Chan [13] as follows: For coating the inside of a 250-ml bottle, 1.0 ml of insecticide at 10 μg/ml of acetone is needed to give a concentration of 10 μg/bottle. For 1000 ml and 0.5- gallon bottles, 4.0 ml of insecticide at 10 μg/ml acetone and 7.57 ml of insecticide at 10 μg /ml acetone are needed, respectively [33]. To compensate and maintain an equivalence of X μg insecticide/bottle, 2.0 ml of X μg insecticide was used to coat the inside of 500 ml bottle. The bottles were coated with insecticide by swirling the acetone:insecticide solution on the bottom, on the sides and on the lid. The bottle was then placed on a mechanical roller for 30 min to dry. During this time the lids were slowly loosened to allow the acetone to evaporate. After 30 min, the caps were removed, and the bottles were rolled until all the acetone had evaporated. The bottles were then left open to dry overnight. For each test replicate, one bottle serving as control was coated with 2.0 ml of acetone. All bottles were reused throughout the experiment after proper cleaning following the procedure described in [33].

**Table 1. Concentrations of insecticides used for the exposure to sandflies.**

| Insecticides | Concentration (μg/ml) | Exposure time |
|---|---|---|
| Alpha-cypermethrin (SP) | 0.25, 0.5, 0.75, 1, 1.5, 2, 3, 4, 5, 8, 10, 25 | 30 min. |
| Deltamethrin (SP) | 0.01, 0.1, 0.5, 0.75, 1, 1.5, 2, 2.5, 3, 4, 5, 8, 10 | 30 min. |
| Malathion (OP) | 0.1, 0.5, 1, 1.5, 2, 2.5, 3, 4, 5, 10, 25 | 30 min. |
| Chlorpyrifos (OP) | 0.01, 0.1, 0.25, 0.5, 0.6, 0.7, 0.8, 1, 2, 3, 4, 5, 6, 7, 8, 10 | 30 min. |
| DDT (OC) | 0.5, 1, 2.5, 5, 10, 25, 50, 75, 100, 150, 200, 300, 500 | 120 min. |

### Insecticide exposure tests and survival curves

**For susceptibility.** The day after the bottles were prepared with insecticide, 20 to 30 sand flies (unfed females and males) at least 2 to 3 days old were aspirated from the colony and gently blown into each bottle. Approximately the same numbers of flies irrespective of sex were utilized for each insecticide-coated bottle, including the control bottle. A minimum of three replicates were completed for each insecticide concentration. Standard exposure time was maintained to 30 min for all insecticides except DDT (120 min) because 30 min exposure time was too short as sand fly survival is nearly 100%, so that it was adjusted depending on the expected and actual sand fly survival rate [33]. After insecticide exposure, the sand flies were captured with a mechanical aspirator and released into a 1-pint cardboard container with fine mesh screen on top, and maintained in a separate incubator under the same humidity, temperature and food source (cotton ball soaked in 30% sugar solution) as the untreated colony. Twenty-four hours (24 h) after insecticide exposure, mortality was recorded. If mortality in the control bottle was between 5 and 20%, mortality in the experimental bottles of that test group were corrected using the Abbott's formula. The mortality correction was not used for the group if the mortality in the control bottle was <5%. If the control mortality exceeded 20%, the entire test group was discarded [37]. Dose-response survival curves were produced and logistic regression models utilized to generate the $LC_{50}$, $LC_{90}$ and $LC_{95}$ for each insecticide using the QCal software [38].

**For diagnostic dose and time.** One day after the bottles were prepared with insecticide, 20 to 30 sand flies (unfed females and males) at least 2 to 3 days old were aspirated from the colony and gently blown into each bottle. Approximately the same numbers of flies, irrespective of sex, were utilized for each insecticide- coated and in the control bottle [13, 35]. Sand flies were aspirated into the control bottle first, then into the three insecticide-coated bottles. Once the flies were aspirated into bottle, the timer was started and the start time recorded as "time zero". Separate timer used for each bottle to maintain accuracy. At time zero, the total number of flies in each bottle was recorded. As the bottle was gently rotated, knockdown mortality (time-to-knockdown) during the exposure test was recorded at 0, 10, 20, 30, 40, 50, 60, 70, 80, 90, 100, 110, and 120 minutes to generate the survival curve, as well as recording mortality after a 24-h of recovery period (24-h mortality) [13]. After completion of exposure time, the same procedure for 24 h recovery periods were followed as described in the susceptibility analysis procedure. If all sand flies were counted as dead before 120 minutes, the flies were kept in the bottle and observed until the 120-minute time point was reached. By plotting time on the X-axis and percent mortality on the Y-axis, time-response survival curves were made for each insecticide. For each insecticide dose, the percent mortality at each time point is the average mortality of all three insecticide-treated bottles. To test the susceptibility status of any vector species against any insecticide using the CDC bottle bioassay, a diagnostic dose and diagnostic time are needed for that insecticide in that region [13]. A diagnostic dose is the lowest dose of insecticide that gives 100% mortality in a susceptible population within a given time period (30–60 minute). If 100% mortality is achieved before this 30–60 minute window, it is understood that the concentration is too high and can lead to masking of resistance. On the other hand, if 100% mortality is achieved after 60 minutes, the concentration is too low to kill all susceptible flies, providing a false-positive result for resistance [13].

## Results

### Survival curve

A dose-response survival regression analysis was performed to estimate $LC_{50}$, $LC_{90}$ and $LC_{95}$ for all five insecticides. Fig 1 shows the survival curves of all five insecticides (alpha-

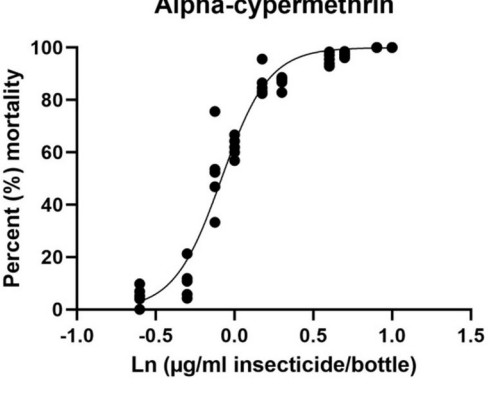

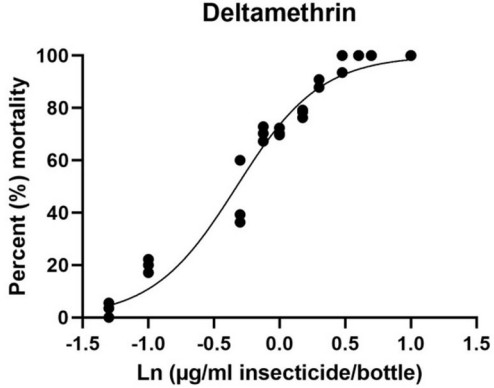

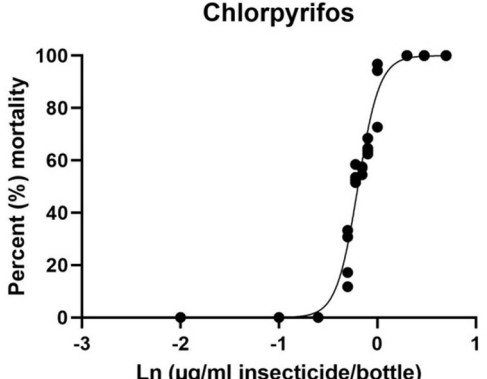

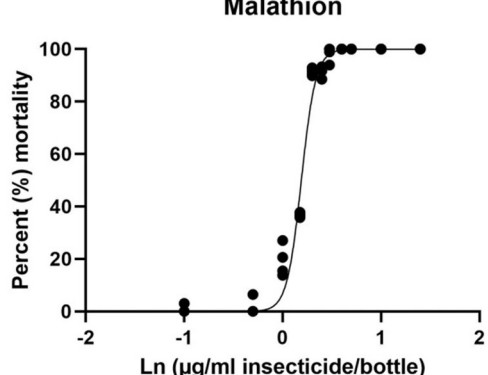

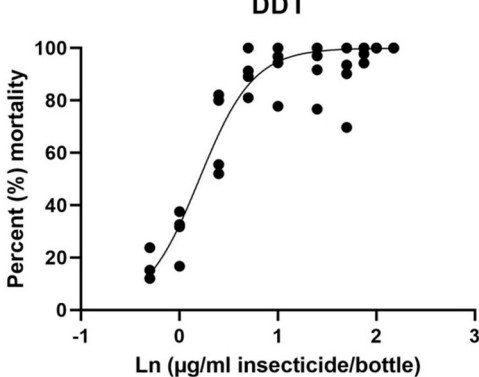

**Fig 1. Dose-response survival curves of five insecticides; Alpha-cypermethrin; Deltamethrin; Malathion; Chlorpyrifos; and DDT.**

**Table 2. Lethal concentration (LC) values causing 50, 90, and 95% mortality (with their respective lower and upper confidence limits) of *P. argentipes*, using the CDC bottle bioassay.**

| Insecticide | LC50 (µg/ml insecticide/bottle) [LL, UL] | LC90 (µg/ml insecticide/bottle) [LL, UL] | LC95 (µg/ml insecticide/bottle) [LL, UL] |
|---|---|---|---|
| Alpha-cypermethrin | 0.830 [0.783, 0.882] | 2.102 [1.881, 2.351] | 2.883 [2.506, 3.317] |
| Deltamethrin | 0.406 [0.353, 0.455] | 2.059 [1.748, 2.425] | 3.595 [2.918, 4.428] |
| Malathion | 1.527 [1.425, 1.636] | 3.072 [2.789, 3.399] | 3.897 [3.443, 4.411] |
| Chlorpyrifos | 0.686 [0.658, 0.715] | 1.080 [0.996, 1.174] | 1.260 [1.138, 1.395] |
| DDT | 1.315 [1.095, 1.579] | 12.562[9.781, 16.135] | 27.061[19.628, 37.311] |

cypermethrin, deltamethrin, chlorpyrifos, malathion and DDT). Table 2 shows the QCal logistic regression parameters and extrapolated $LC_{50}$, $LC_{90}$ and $LC_{95}$ for each insecticide, with their respective lower and upper confidence limits. The $LC_{50}$ of malathion (1.527 µg/ml) is comparatively higher than DDT 1.315 µg/ml, alpha-cypermethrin 0.83 µg/ml, chlorpyrifos 0.686 µg/ml and Deltamethrin 0.406 µg/ml. The $LC_{95}$ was substantially greater than $LC_{90}$ for alpha-cypermethrin, deltamethrin, chlorpyrifos and malathion, while it was more than two fold higher for DDT ($LC_{90}$-12.56 µg/ml; $LC_{95}$-27.06 µg/ml).

## Diagnostic dose and time

A time-response survival curve for each insecticide was created by plotting the time on the X-axis and percent mortality on the Y-axis. For all, the time-to-knockdown survival curves, the time to reach 100% mortality, decreases with increasing insecticide concentration. Two diagnostic doses for each insecticide were determined. Diagnostic doses and times to knockdown and a diagnostic dose after 24-h mortality for all five insecticides are presented in Table 3. Representative survival curves are shown in Fig 2. The diagnostic doses to knockdown mortality between stipulated times of 30 and 60 minutes were similar to the $LC_{95}$ value for alpha-cypermethrin, deltamethrin and malathion, and was comparatively higher for chlorpyrifos.

**Pyrethroids.** Two time-to-knockdown diagnostic doses for alpha-cypermethrin were determined at 2.5µg/ml for 50 minutes and 3µg/ml for 40 minutes. Deltamethrin had a diagnostic dose of 4µg/ml at 60 minutes and 5µg/ml at 50 minutes. The time-to-knockdown diagnostic doses of deltamethrin was relatively higher, as compared to alpha-cypermethrin, but diagnostic dose after 24-h mortality was similar for both the insecticides.

**Organophosphates.** As compared to the LC values of malathion and chlorpyrifos, time-to-knockdown diagnostic doses were higher for both the insecticides. Two time-to-

**Table 3. Diagnostic doses and diagnostic times for insecticides at time-to-knockdown and mortality after 24-hours.**

| Insecticides | Diagnostic dose | Diagnostic time-to-knockdown | Diagnostic dose after 24-hour mortality |
|---|---|---|---|
| Alpha-cypermethrin | 2.5 µg/ml | 50 min | 2 µg/ml |
| | 3 µg/ml | 40 min | |
| Deltamethrin | 4 µg/ml | 60 min | 2 µg/ml |
| | 5 µg/ml | 50 min | |
| Malathion | 4 µg/ml | 60 min | 1 µg/ml |
| | 5 µg/ml | 50 min | |
| Chlorpyrifos | 5 µg/ml | 60 min | 2 µg/ml |
| | 6 µg/ml | 50 min | |
| DDT | 12 µg/ml | 60 min | 25 µg/ml |
| | 15 µg/ml | 50 min | |

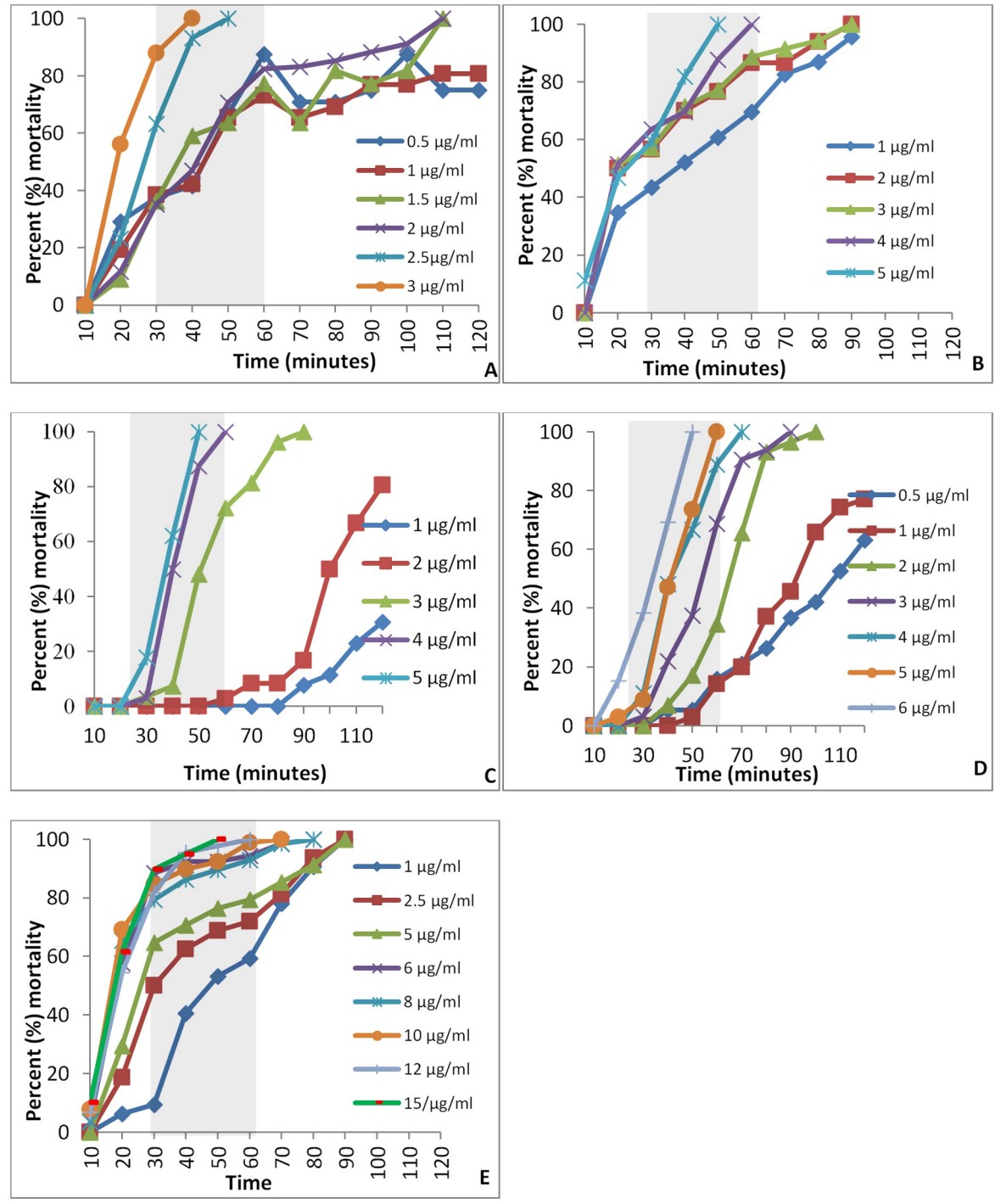

**Fig 2. Time-to-knockdown survival curves for five insecticides; A, Alpha-cypermethrin; B, Deltamethrin; C, Malathion; D, Chlorpyrifos; E, DDT.**

knockdown diagnostic doses for malathion were determined to be 4 μg/ml at 60 minutes and 5 μg/ml at 50 minutes. For chlorpyrifos, the diagnostic dose were 5μg/ml at 60 minutes and 6μg/ml at 50 minutes, and 24-h mortality diagnostic doses were 1 μg/ml and 2 μg/ml, respectively.

**Organochlorine.** The time-to-knockdown diagnostic doses and 24-h-mortality diagnostic doses for DDT were nearly similar to the values of $LC_{90}$ and $LC_{95}$, respectively. Two time-to-knockdown diagnostic doses were 12μg/ml at 60 minutes and 15μg/ml at 50 minutes, and its 24-h mortality diagnostic dose was 25μg/ml.

## Discussion

This study shows that different insecticides have different lethal concentrations and times. Malathion causes delayed mortality, as compared to chlorpyrifos, deltamethrin and alpha-cypermethrin, which is similar to the results of Denlinger et al for *P. papatasi* [33]. Among the five insecticides tested, sand flies were least susceptible to DDT exposure, even with an exposure time of 120 minutes. Unlike pyrethroids, which inhibit action-potential propagation of the sodium channels involved in the central nervous system and in the peripheral nervous system [39], DDT is known to block mainly the sodium channels in the peripheral nervous system [39]. Affecting only the peripheral nervous system requires more time and higher doses to cause excitatory paralysis leading to death [33,39]. Similar observations were reported in the insecticide susceptibilities of *Phlebotomus perniciosus* and *P. papatasi* against DDT [37,40].

Historically, DDT had been used worldwide to control sand flies by direct or indirect interventions. Published reports also describe sandflies being susceptible, tolerant or resistance to DDT from India, Nepal, Iran and Turkey (WHO, 1986, [27,41–44]. The data from this study also suggest that large doses of DDT are required, which may produce strong selection pressure for resistance if applied injudiciously [45]. Compounded with years of DDT use, field populations of sand flies may be able to develop resistance to DDT more quickly than to any other insecticides. Due to widespread resistance of *P. argentipes* to DDT in India, IRS with DDT was replaced with alpha-cypermethrin in 2016. Even after 3–4 years of complete withdrawal, *P. argentipes* resistance to DDT was not reversed [44]. Even though this is a short time frame to notice any change in DDT resistance, another study in India also showed non-reversible of DDT resistance in mosquitoes even 30 years after stopping IRS with DDT [46].

Alpha-cypermethrin and deltamethrin were found to be more effective at lower concentrations (Fig 1 and Table 2). These results support the previous findings of others [27,44,47,48,49]. Both insecticides belong to the type II pyrethroids, which cause sodium channel modifications that can last for many seconds and are better at causing mortality in insects at low concentrations [39]. The low lethal values at $LC_{50}$, $LC_{90}$ and $LC_{95}$ support previous research and are consistent with physiological differences between two pyrethroid groups [50,51]. *Phlebotomus argentipes* was highly susceptible to pyrethroids [44,49] and field trials in India, Bangladesh and Nepal have shown high entomological efficacy of IRS with alpha-cypermethrin or deltamethrin [52–54].

The lethal concentration of chlorpyrifos in laboratory colony insecticide-susceptible sand flies suggests high susceptibility. These results are similar to the susceptibilities of *P. papatasi* and *Lutzomyia longipalpis* to chlorpyrifos [33]. When converted to μg malathion/ml for comparison, the *P. argentipes* laboratory colony had a $LC_{90}$ of 3.07 μg/ml and a $LC_{95}$ of 3.89 μg/ml, which were close to the concentrations determined for *L. longipalpis* ($LC_{50}$ of 3.45 μg/ml and $LC_{95}$ of 4.08 μg/ml) [33]. Malathion has not been used in the VL- endemic areas in India for vector control, but its use in agricultural pest control cannot be ruled out [44]. *Phlebotomus argentipes* collected from Puduchery in India and Delft islands of Sri Lanka was reported resistant to malathion [24,55].

Evaluation of the susceptibility status and resistance detection of sand flies has been hampered by a lack of validated data on diagnostic doses and times. In absence of sand fly specific WHO susceptibility test procedure and impregnated papers, the WHO diagnostic doses for

malaria vectors are used for resistance monitoring [56]. Diagnostic doses for *Anopheles* insecticide resistance are likely to be higher than those for sand flies, because sand flies are likely to fly less than mosquitoes in bioassays and spend more time in contact with the substrate [56]. The present study established diagnostic doses or concentrations and times for different insecticides using the CDC bottle bioassay on *P. argentipes* from India, which will strengthen the collection of diagnostic doses and times available for *Phlebotomus* spp. [26,32,34,35].

There have been very limited studies that have determined the time-dependent dose mortality (diagnostic dose and diagnostic times) for *P. argentipes* using the CDC bottle bioassay. With the results presented in this study, comparisons can now be made for alpha-cypermethrin, deltamethrin and DDT. The KAMRC laboratory *P. argentipes* colony shows 100% mortality in 40 minutes using 3μg/ml alpha-cypermethrin, while Anderson (2020) [57] determined a concentration of 3μg/ml caused 100% mortality in 45 minutes. The *P. argentipes* lab colony also required 5 μg/ml deltamethrin to cause 100% mortality in 50 minutes, which was nearly similar to the diagnostic dose and diagnostic times of *P. papatasi* and *L. longipalpis* (5 μg/ml in 60 minutes) [35]. When comparing the diagnostic dose and diagnostic times of DDT against *P. argentipes*, it was 15μg/ml for 50 minutes. This concentration is almost double the diagnostic doses and diagnostic times of *P. papatasi* and *L. longipalpis* (7.5μg/ml for 30 minutes) [35]. One possible explanation is that Indian sand fly populations are highly resistant to DDT due to intense historical exposure.

The potential limitation of this study is that we used an established laboratory adapted strain of *P. argentipes*. Lethal concentration and lethal time from insecticide susceptible laboratory and wild caught sand flies may differ because of highly variable natural conditions. Wild populations may exhibit different behaviors, physiologies, longevity and developmental time that make them more or less susceptible to insecticides [33]. The baseline lethal-concentration values and diagnostic doses and times for each insecticide for a susceptible population of a vector species from a specific geographic area are fundamentally required when assessing resistance in field populations [58–60]. Similarly, the diagnostic concentration and times presented in this study should be used as an initial reference dose and time for assessing resistance in field populations, as well as for determining the diagnostic doses and diagnostic times for other insecticide-susceptible sand fly populations, and also provide valuable base line data for developing CDC bottle bioassay intensity rapid diagnostic test (I-RDT's) for sand flies. Further studies using CDC bottle bioassay are still needed from various field populations to have comparable diagnostic-time data. These studies will be useful for evaluating comparability and validating diagnostic doses between different populations of sand fly and species.

As India enters into the post-elimination phase of VL, it will be critical to ensure that resistance does not develop significantly. The base line data determined here specific to insecticide susceptibility and diagnostic doses and times can be used to monitor susceptibility status of various wild populations of sand fly and species from different geographic areas. Understanding insecticide susceptibility and resistance in communities is key to reducing potential resurgence of sand fly populations and potential spread of the parasite. Being able to predict insecticide resistance will allow for professionals to make adjustments to current indoor residual spraying methods, identify communities of concern, and propose changes or adjustments, when needed, to insecticides currently being used.

## Acknowledgments

We acknowledge Mr. Anil Sharma and all the laboratory staff at the KAMRC for administrative and logistical support.

## Author Contributions

**Conceptualization:** Rahul Chaubey, Anurag Kumar Kushwaha, Puja Tiwary, Shawna Hennings, Edgar Rowton, Christine A. Petersen, Scott A. Bernhardt.

**Data curation:** Rahul Chaubey, Ashish Shukla, Anurag Kumar Kushwaha, Puja Tiwary, Shakti Kumar Singh, Shawna Hennings, Edgar Rowton, Scott A. Bernhardt.

**Formal analysis:** Rahul Chaubey, Ashish Shukla, Scott A. Bernhardt.

**Funding acquisition:** Shyam Sundar.

**Methodology:** Rahul Chaubey, Anurag Kumar Kushwaha, Puja Tiwary, Shawna Hennings, Om Praksh Singh, Phillip Lawyer, Edgar Rowton, Christine A. Petersen, Scott A. Bernhardt.

**Project administration:** Scott A. Bernhardt, Shyam Sundar.

**Resources:** Shyam Sundar.

**Supervision:** Scott A. Bernhardt.

**Validation:** Scott A. Bernhardt.

**Writing – original draft:** Rahul Chaubey, Scott A. Bernhardt.

**Writing – review & editing:** Phillip Lawyer, Christine A. Petersen, Scott A. Bernhardt.

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
