## [Decision Letter · Decision Letter 0]

27 Dec 2022

Dear Dr. Bernhardt,

Thank you very much for submitting your manuscript "Assessing insecticide susceptibility, diagnostic dose and time for the sand fly Phlebotomus argentipes, the vector of visceral leishmaniasis in India, using the CDC bottle bioassay" for consideration at PLOS Neglected Tropical Diseases. As with all papers reviewed by the journal, your manuscript was reviewed by members of the editorial board and by several independent reviewers. In light of the reviews (below this email), we would like to invite the resubmission of a significantly-revised version that takes into account the reviewers' comments. 

Although all three reviewers acknowledged the value of the work and its importance for the field, they also pointed out several modifications to be performed before the manuscript can be accepted for publication in PLOS NTD.

Major modifications:

- please correct all typo errors (word spacing, section titles, bibliography formatting, ...).

- As the data presented may be used as a baseline by subsequent studies, the insecticide susceptibility profile of the laboratory strain used in the present study should be made cleared in the method section (referecne #36 does not report resistance data/status) and supported by ANY available data. Defining insecticide diagnostic doses usually implies testing multiple 'susceptible' strains of distinct geographical origins to be sure that the diagnostic dose proposed is coherent. This point is crucial important as the presence of DDT (and PYR?) resistance allele is highly suspected in the line used for the present study... This should also be further discussed in the discussion section (e.g. comparison with other studies).

- In the results section, I am not sure it is necessary to make a whole paragraphe comparing LC50s from different insecticides as different molecules have different inherent toxicity.

Minor modifications:

- Line 215: the definition of "diagnostic dose" proposed is not the one used by WHO (which is usually twice the dose killing 100% of susceptible individuals).

- line 320: I do not agree about the statement of different biochemical target between PYR and DDT, modify or provide a solide reference to support this statement.

- line 378-380: tone down. before being used as a reference, the doses and times reported in the present study should be compared with data obtained from other lines of the same species.

In addition, please ensure to take into account all other corrections asked by reviewers.

We cannot make any decision about publication until we have seen the revised manuscript and your response to the reviewers' comments. Your revised manuscript is also likely to be sent to reviewers for further evaluation.

Sincerely,

Jean-philippe David

Academic Editor

Alvaro Acosta-Serrano

Section Editor

Reviewer's Responses to Questions

Key Review Criteria Required for Acceptance?

Methods

-Are the objectives of the study clearly articulated with a clear testable hypothesis stated?

-Is the study design appropriate to address the stated objectives?

-Is the population clearly described and appropriate for the hypothesis being tested?

-Is the sample size sufficient to ensure adequate power to address the hypothesis being tested?

-Were correct statistical analysis used to support conclusions?

-Are there concerns about ethical or regulatory requirements being met?

Reviewer #1: -Yes, objectives are clearly identified.

-Yes

-Yes

-Yes

-Yes

Reviewer #2: All issues are addressed except ethical concern, even not mentioned anything about the regulatory authority who had approved the proposal for conducting the study.

Reviewer #3: Yes in large part. The numbers of sand flies tested should be given in the abstract.

Results

-Does the analysis presented match the analysis plan?

-Are the results clearly and completely presented?

-Are the figures (Tables, Images) of sufficient quality for clarity?

Reviewer #1: -Yes

-Yes all results were clearly presented

-Yes

Reviewer #2: Acceptable.

Reviewer #3: Though the methods section says the experiments were repeated 3 times, that information isn't shown in the results section. 

None of the figures or tables have legends, therefore difficult to interpret.

The abbreviated terms used in the text and in illustrations should be described properly e.g. 24 hour mortality

Conclusions

-Are the conclusions supported by the data presented?

-Are the limitations of analysis clearly described?

-Do the authors discuss how these data can be helpful to advance our understanding of the topic under study?

-Is public health relevance addressed?

Reviewer #1: -Yes, supported

-No limitation was noted

-Yes

-Yes

Reviewer #2: Not drawn any appropriate conclusion. 

Not discussed the strength and weakness of the study.

Not mentioned the public health relevance of the study.

Reviewer #3: Yes, in large part.

However, the conclusions could be toned down and viewed against the existing information from the region. It'll be useful to access more recent publications on insecticide susceptibility studies in the South Asian region to discuss the results more meaningfully. 

Limitations of the studies also could be added.

Editorial and Data Presentation Modifications?

Reviewer #1: (No Response)

Reviewer #2: Major revision is needed.

Reviewer #3: There are a few typographical and grammatical errors as well that should be corrected.

Summary and General Comments

Reviewer #1: Manuscript submitted by Chaubey et al. is is describing a the resistance profile and experiments about insecticide. The subject of this manuscript is within the field of interest of the Journal and the methods are appropriate and well described. The paper is well written and concise and, as such, is suitable for publication in PNTD. The literature is well reviewed, experimental design and statistical methods are adequate. There are some specific points recommendations.

Line 50: correct "insectcidesare"

Line 51: correct " Sand flieshad"

Line 56: correct "timesare"

Line 68: correct "Leishmaniadonavani"

line 73: correct "Thisresting"

Line 79: correct "leishmaniasisresurged"

There are also other typos that authors should check trough the manuscript.

Line 180: I dont understand why specimens were released to bottles with mixed genders? Why both sexes did not tested separately?

Line 185: I think there should be a standard exposure time? Why its 30 mins or 120 mins.? Also, did 30 mins results and 120 mins results were analyzed together?

Reviewer #2: Mechanical aspirator used for collection of sandflies that causes the damage of sandflies that may enhanced the mortality of the exposed sandflies.

Only one timer used for four bottles, but for accurate time maintenance four timers should use.

Line 261-262: Reference should not mention in the result, however it may move to 'discussion' section.

Line 312: Insecticides tested against sandflies, but mentioned different disease vectors, the statement is confusing.

Line 310-315: Should be deleted.

Line 339: What is Deb et al. 2021?

How findings of the study will facilitate the "Indian kala-azar elimination program" need to mention at the end of the manuscript (conclusion).

Reviewer #3: It's a useful study that describes the insecticide susceptibility pattern of sand flies in a laboratory-bred colony in India, which will provide useful information by way of baseline data, which is lacking. 

The tables and figures should contact legends to enable easy interpretation by the readers and abbreviated terms should be spelt out.

More recent publications on insecticide susceptibility in the region should be added to the discussion.

Reviewer #1: Mehmet Karakus

Reviewer #2: confidential

Reviewer #3: confidential
---

## [Decision Letter · Decision Letter 1]

31 Mar 2023

Dear Dr. Bernhardt and co-authors,

We are pleased to inform you that your manuscript 'Assessing insecticide susceptibility, diagnostic dose and time for the sand fly Phlebotomus argentipes, the vector of visceral leishmaniasis in India, using the CDC bottle bioassay' has been provisionally accepted for publication in PLOS Neglected Tropical Diseases.

Best regards,

Jean-Philippe David

Academic Editor

Álvaro Acosta-Serrano

Section Editor

Dear authors,

Considering the changes made to the manuscript and your answers to reviewers, your manuscript is now suitable for publication in Plos NTD. However, the MS may still requires some minor formatting changes (typos, grammar, figure format) that will be handled by the production team (proof stage).

You will find below a few remaining comments from reviewers that may deserve further attention to the formatting of the manuscript before publication.

Best regards,

Jean-Philippe David

<style type="text/css">p.p1 {margin: 0.0px 0.0px 0.0px 0.0px; line-height: 16.0px; font: 14.0px Arial; color: #323333; -webkit-text-stroke: #323333}span.s1 {font-kerning: none

</style>

Reviewer's Responses to Questions

**Key Review Criteria Required for Acceptance?**

**Methods**

-Are the objectives of the study clearly articulated with a clear testable hypothesis stated?

-Is the study design appropriate to address the stated objectives?

-Is the population clearly described and appropriate for the hypothesis being tested?

-Is the sample size sufficient to ensure adequate power to address the hypothesis being tested?

-Were correct statistical analysis used to support conclusions?

-Are there concerns about ethical or regulatory requirements being met?

Reviewer #1: Yes

Reviewer #2: Addressed properly.

**Results**

-Does the analysis presented match the analysis plan?

-Are the results clearly and completely presented?

-Are the figures (Tables, Images) of sufficient quality for clarity?

Reviewer #1: Yes

Reviewer #2: Figure-2 need to revisit to make sure the correct presentation.

**Conclusions**

-Are the conclusions supported by the data presented?

-Are the limitations of analysis clearly described?

-Do the authors discuss how these data can be helpful to advance our understanding of the topic under study?

-Is public health relevance addressed?

Reviewer #1: Yes

Reviewer #2: Are ok.

**Editorial and Data Presentation Modifications?**

Reviewer #1: Accept

Reviewer #2: Minor Revision.

**Summary and General Comments**

Reviewer #1: All corrections were made. I dont suggest minor revision nor acceptance. I want to proceed without recommendation

Reviewer #2: Not clearly addressed.

PLOS authors have the option to publish the peer review history of their article (what does this mean?). If published, this will include your full peer review and any attached files.

Reviewer #1: **Yes: **Mehmet Karakus

Reviewer #2: No

---

## [Editor Report · Acceptance letter]

28 Apr 2023

Dear Dr. Bernhardt,

We are delighted to inform you that your manuscript, "Assessing insecticide susceptibility, diagnostic dose and time for the sand fly Phlebotomus argentipes, the vector of visceral leishmaniasis in India, using the CDC bottle bioassay," has been formally accepted for publication in PLOS Neglected Tropical Diseases.

Best regards,

Shaden Kamhawi

co-Editor-in-Chief

Paul Brindley

co-Editor-in-Chief
